# A Combined GLP-1/PPARa/CB1-Based Therapy to Restore the Central and Peripheral Metabolic Dysregulation Induced by a High-Fructose High-Fat Diet

**DOI:** 10.3390/ijms26062420

**Published:** 2025-03-07

**Authors:** Marialuisa de Ceglia, Nabila Rasheed, Rubén Tovar, Inés Pareja-Cerbán, Andrea Arias-Sáez, Ana Gavito, Silvana Gaetani, Carlo Cifani, Fernando Rodríguez de Fonseca, Juan Decara

**Affiliations:** 1Grupo de Neuropsicofarmacología, Instituto IBIMA-Plataforma BIONAND, Unidad de Gestión Clínica de Salud Mental, Hospital Regional Universitario de Málaga, Av. de Carlos Haya, 29010 Málaga, Spain; rubentovar14@gmail.com (R.T.); ines.pareja@externos.isciii.es (I.P.-C.); ariassaezandrea99@gmail.com (A.A.-S.); analugavito@hotmail.com (A.G.); fernando.rodriguez@ibima.eu (F.R.d.F.); 2School of Pharmacy, Pharmacology Unit, University of Camerino, 62032 Camerino, Italy; carlo.cifani@unicam.it; 3Department of Physiology and Pharmacology “V. Erspamer”, Sapienza University of Rome, 00185 Rome, Italy; nabila.rasheed@uniroma1.it (N.R.); silvana.gaetani@uniroma1.it (S.G.)

**Keywords:** diet, obesity, insulin pathway, brain, tau phosphorylation, GLP-1, endocannabinoid system

## Abstract

Obesity remains a major epidemic in developed countries, with a limited range of effective pharmacological treatments. The pharmacological modulation of PPARα, CB1, or GLP-1 receptor activity has demonstrated beneficial effects, including anti-obesity actions. In this study, we evaluated a novel amide derivative of oleic acid and tyrosol (Oleyl hydroxytyrosol ether, OLHHA), a PPARα agonist, and CB1 antagonist, in combination with the GLP-1 agonist liraglutide (LIG), as an effective multitarget therapy to improve both the peripheral and central alterations in an animal model of diet-induced obesity. In rats, exposure to a high-fat high-fructose diet (HFHFD) induced weight gain and increased plasma triglycerides, LDL, and hepatic parameters. In the brain, the HFHFD provoked disruptions in the expression of proteins regulating food intake, the endocannabinoid system, the insulin pathway, and inflammation and resulted in altered tau expression and phosphorylation, thus indicating neurodegenerative changes. Based on our results, the administration of LIG or OLHHA alone was insufficient to completely reverse the alterations noticed at the peripheral and central levels. On the other hand, the combined treatment with both compounds (OLHHA+LIG) was the most effective in promoting body weight loss and ameliorating both the central and peripheral alterations induced by HFHFDs in rats. This multitarget therapeutic approach could represent a promising strategy for treating obesity and associated comorbidities.

## 1. Introduction

In recent years, obesity has been recognized as an increasingly prevalent pathology. Reports of the World Obesity Atlas estimate that in 2020, more than 2.6 billion people were overweight or obese, covering 38% of the world’s population. This percentage is predicted to reach 51% by 2035 [1]. 

Obesity is defined as a chronic multifactorial disease characterized by excessive fat deposits and is diagnosed primarily when the body mass index (BMI—measured as weight (kg)/height (m)^2^) is equal to or greater than 30. However, additional measurements of waist circumference and abdominal fat are necessary for a more accurate assessment [2]. The most common causes of obesity are overeating, the consumption of hypercaloric food rich in sugar and fat, and a sedentary lifestyle. However, its development can also be influenced by genetic and socioeconomic factors or other diseases.

Obesity is associated with numerous comorbidities, which involve several organs and systems of the body [3,4], with high cardiovascular risk-associated hyperlipidemia and hepatic damage being the most frequent. Most importantly, recent studies demonstrate that obesity is a risk factor for the onset of neurodegenerative diseases and cognitive decline. Obese patients are likely to develop Alzheimer’s disease (AD) and vascular dementia in later life, along with an accelerated trajectory of cognitive aging, suggesting that obesity can have an immediate detrimental impact on cognitive functioning [5]. Other neurodegenerative diseases which have been associated with obesity are multiple sclerosis and Parkinson’s disease [6].

The link between obesity and neurodegenerative diseases depends on numerous factors. Firstly, obesity is characterized by low-grade chronic inflammation driven by the altered release of pro-inflammatory cytokines and activation of the nuclear factor k-light-chain-enhancer of activated B cells (NF-kB). This inflammatory state exacerbates insulin resistance, leptin resistance, and adipose tissue expansion [6]. Chronic low-grade inflammation has severe consequences in the central nervous system, where it disrupts the permeability of the blood/brain barrier (BBB), thereby facilitating leukocyte infiltration and promoting neurodegenerative diseases associated with impaired synaptic plasticity and neurogenesis. These processes also result in alterations in brain metabolism [6].

Obesity-induced insulin resistance is a crucial link between this condition and the insurgence of neurodegenerative diseases, in particular AD. Insulin exerts neuroprotective actions and contributes to neuronal and synaptic formation, development, and repair [5]. Impaired brain insulin responsiveness contributes to the formation of beta-amyloid plaques and tubulin-associated unit (tau) hyperphosphorylated aggregates [7], both hallmarks of AD’s pathology. Insulin resistance in the brain may also exacerbate oxidative stress and inflammation, further driving neurodegeneration [8]. For these reasons, and based on the existing literature linking diabetes and AD, Alzheimer’s disease is now regarded as type 3 diabetes.

Moreover, obesity has been associated with the profound dysregulation of the endocannabinoid system (ECS), a key homeostatic signaling system at both peripheral and central levels. For this reason, drugs acting on the ECS have been tested for the treatment of this pathology [9]. Endocannabinoids 2-acylglicerol (2-AG) and anandamide (AEA) exert neuroprotective actions in the brain through the activation of cannabinoid receptor type 1 and 2 (CB1, CB2) and peroxisome proliferator-activated receptor alpha (PPARα), by promoting anti-inflammatory mechanisms and attenuating BBB permeability and excitotoxicity [10,11]. Alterations in the ECS have been detected in numerous neurodegenerative illnesses, suggesting that it has a pivotal role in the mechanisms developing and sustaining these pathologies [11]. This suggests that the ECS could be a link between obesity and neurodegenerative diseases.

Seeing these premises, it appears clear that the treatment of obesity is complex and that the medications proposed for this condition must be directed not only at body weight (BW) loss but also address peripheral comorbidities (hyperlipidemia, metabolic syndrome, and cardiac and hepatic issues) and act at the brain level, exerting a neuroprotective effect. In this direction, molecules acting on the glucagon-like peptide receptor 1 (GLP-1R) have gained particular interest since their administration resulted in consistent BW loss [12], a positive effect on cardiac function [13], and the management of diabetes and metabolic syndrome while also exerting neuroprotective effects [14]. Among the GLP1-R agonists, one of the most prescribed is liraglutide (LIG), which is used for treating type 2 diabetes and obesity. LIG administration enhances insulin secretion from the pancreas, slows gastric emptying, reduces hepatic glucose production, and stimulates satiety signals in the hypothalamus, leading to improved glycemic control and BW reduction. Additionally, LIG has been associated with protective effects on the liver, cardiovascular system, and brain [15].

In this study, rats were exposed to a high-fat high-fructose diet (HFHFD) to model morbid obesity and metabolic syndrome [16]. BW gain and obesity-related peripheral parameters, including blood lipids and hepatic enzymes, were assessed. Moreover, the effect of HFHFD exposure on brain insulin resistance, tau hyperphosphorylation, the ECS pathway, neuroinflammation, and neurogenesis was evaluated in key brain regions involved in the regulation of metabolism and cognition, such as the hypothalamus (HYPO), hippocampus (HIPPO), and prefrontal cortex (PFC).

In parallel, HFHFD-fed rats were administered submaximal doses of LIG, OLHHA, or a combination of the two drugs to search for possible synergistic effects of the combined therapy. OLHHA is a dual ligand acting as a neutral CB1 antagonist/PPARα agonist previously patented in our laboratory for its hypophagic and hepatoprotective properties [17] and its efficacy in reducing binge-drinking behavior [18]. The final aim was to determine whether these therapies could restore the metabolic, signaling, and inflammatory alterations imposed by a HFHFD and identify the most effective therapeutic strategy.

Finally, correlation analysis was conducted to assess whether obesity-related peripheral parameters (blood lipids and hepatic enzymes) can be associated with brain protein expression, potentially serving as biomarkers of brain health in obese individuals.

## 2. Results

This work focused on two aspects: first, analyzing the diet-induced changes by comparing the standard diet group (CTRL STD) with the HFHFD group (CTRL HFHFD); second, assessing the effects of compound treatments (HFHFD LIG, HFHFD OLHHA, HFHFD OLHHA+LIG) in the HFHFD group to determine which treatment was the most effective in reversing the alterations caused by the diet. 

### 2.1. Body Weight and Plasma Biochemistry

Table 1 shows the effect of the exposure to diet and treatment on BW gain, plasma biochemistry, and markers normally associated with obesity.

Two-way ANOVA evidenced an effect of diet for BW gain, triglycerides, low-density lipoprotein (LDL), and bilirubin. At the end of the experiment, the CTRL HFHFD animals displayed a significant increase in BW gain, triglycerides, LDL, and bilirubin when compared to the CTRL STD animals.

The effect of treatment was observed in two-way ANOVA for BW gain, triglycerides, LDL, AST (aspartate aminotransferase), ALT (alanine aminotransferase), and bilirubin. The treatment with LIG slightly decreased BW gain in HFHFD-fed animals. OLHHA or combined OLHHA+LIG significantly reduced BW gain versus vehicle administration (CTRL HFHFD group). OLHHA+LIG administration could be considered the most effective alternative due to the significant difference between the LIG and OLHHA administration alone. 

Compared to the CTRL HFHFD group, the HFHFD LIG group presented significantly reduced triglycerides and LDL levels but no change in hepatic parameters. On the other hand, OLHHA treatment alone reduced LDL and bilirubin significantly. Most importantly, the combined treatment (OLHHA+LIG) significantly restored normal triglycerides, LDL, AST, and ALT compared to vehicle administration, resulting in the most effective alternative. Significant differences were noticed among the combined treatments and LIG or OLHHA administration alone, suggesting a synergic action of the two compounds. Appendix A shows the complete statistical details of the analysis (p, F, and degrees of freedom).

### 2.2. Protein Expression

#### 2.2.1. Hypothalamus

Figure 1A reports the effects of diet exposure or treatment administration on endocannabinoid system (ECS) expression.

In the HYPO, a significant effect of diet was reported by two-way ANOVA for the expression of CB1, diacylglycerol lipase alpha (DAGLα), diacylglycerol lipase beta (DAGLβ), monoacylglycerol lipase (MAGL), N-acyl phosphatidylethanolamine-phospholipase D (NAPE-PLD), Fatty-acid amide hydrolase (FAAH), and the ratios DAGLα/MAGL, DAGLβ/MAGL, and NAPE-PLD/FAAH. The CTRL HFHFD animals displayed a significant decrease in the expression of CB1, DAGLα, and the ratios DAGLα/MAGL and DAGLβ/MAGL versus the CTRL STD ones. Significant increases in DAGLβ, MAGL, NAPE-PLD, FAAH, and the ratio NAPE-PLD/FAAH were reported in CTRL HFHFD rats compared to CTRL STD rats.

ANOVA showed a significant effect of treatment on the expression of CB2, PPARα, DAGLα, DAGLβ, MAGL, NAPE-PLD, and FAAH.

HFHFD LIG, HFHFD OLHHA, and HFHFD OLHHA+LIG animals showed a significantly decreased expression of PPARα compared to the CTRL HFHFD.

Treatment with OLHHA and OLHHA+LIG significantly reduced CB2 and NAPE-PLD expression in HFHFD-fed animals and significantly normalized the expression of DAGLβ and the ratio NAPE-PLD/FAAH. The combined treatment with OLHHA+LIG significantly normalized MAGL, FAAH, and the ratio DAGLα/MAGL to values comparable to the CTRL STD group.

Figure 1B reports the effects of diet exposure or treatment administration on proteins regulating food intake.

A significant effect of the diet was evidenced by two-way ANOVA in GLP1-R and neuropeptide Y receptor type 1 (NPY1-R) expression. CTRL HFHFD animals displayed a significant increase in GLP-1R and NPY1-R, along with a significant decrease in the expression of the MOR receptor, compared to CTRL STD rats.

ANOVA displayed a significant effect of treatment in the expression of GLP1-R, MOR, and NPY1-R. The combined treatment with OLHHA+LIG significantly increased GLP1-R expression and reduced NPY1-R compared to the CTRL HFHFD group. No significant variation was elicited by LIG or OLHHA treatment in the expression of these proteins.

Figure 2A shows the effects of diet exposure or treatment administration on the insulin pathway and protein activation/inhibition, evaluated by phosphorylation.

A significant effect of diet was observed in two-way ANOVA for the expression of phosphoinositide 3-kinase (PI3K) and its phosphorylated form and glycogen synthase kinase beta (GSK3β) and its phosphorylated form. Compared to the CTRL STD animals, CTRL HFHFD animals displayed a significant decrease in the expression of total PI3K and GSK3β and a significant increase in the phosphorylation of both proteins.

A significant effect of treatment was observed in two-way ANOVA for the expression of insulin receptor beta (IRβ), insulin receptor substrate 1 (IRS-1) total and phosphorylated form in tyrosine (TYR), and protein kinase B (AKT). Compared to the CTRL HFHFD group, treatment with LIG elicited no significant change. The HFHFD OLHHA group showed a significant decrease in the total expression of IRS-1 compared to the control group (CTRL HFHFD). Conversely, HFHFD OLHHA+LIG rats displayed a significant reduction in the expression of IRβ and phosphorylated form of IRS-1 (TYR), and a significant increase in AKT total expression compared to all treatments (CTRL HFHFD, HFHFD LIG, and HFHFD OLHHA). Moreover, a significant reduction in the total expression of IRS-1 and a significant increase in phosphorylated PI3K was noticed in HFHFD OLHHA+LIG rats versus CTRL HFHFD rats. 

Figure 2B shows the effects of diet exposure or treatment administration on metabolic sensors and protein activation/inhibition, evaluated by phosphorylation.

ANOVA evidenced a significant effect of diet for the expression of the mammalian target of rapamycin (mTOR), phosphorylated form of Extracellular Signal-Regulated Kinase 1 (ERK1), and total and phosphorylated form of Extracellular Signal-Regulated Kinase 2 (ERK2). CTRL HFHFD animals showed a significant decrease in the expression of total mTOR, phosphorylated form of ERK1, and total ERK2, and a significant increase in the phosphorylated form of ERK2 compared to the CTRL STD.

Also, a significant effect of treatment was identified by ANOVA for the total and phosphorylated form of mTOR and the total and phosphorylated form of ERK2. The treatment with LIG, OLHHA, and OLHHA+LIG significantly increased total ERK2 and reduced the expression of its phosphorylated form versus vehicle administration. Furthermore, the HFHFD OLHHA+LIG group reported an increased expression in the total form of mTOR and ERK1 compared to the CTRL HFHFD.

Figure 3A shows the variations elicited by diet or treatment in total tau, its phosphorylated forms, AT8 (pSer202/pThr205) and AT100 (pThr212/pS214), and the expression of a kinase phosphorylating tau, cyclin-dependent kinase 5 (CDK5).

A significant effect of the diet was shown in the ANOVA for the expression of CDK5 and its receptor subunits, p35 and p25. Compared to naïve animals, the CTRL HFHFD displayed an increase in total tau and significant alterations in CDK5 (decrease), p35, and p25 (increase).

Treatment elicited a significant effect in the expression of total tau, p35, and p25.

OLHHA and OLHHA+LIG administration in HFHFD-fed rats significantly decreased p35 expression versus vehicle administration. The combined treatment with OLHHA+LIG significantly restored p25 and total tau expression when compared to vehicle, LIG, or OLHHA administration.

Figure 3B shows variations elicited by diet or treatment in the markers of trophism, neuroinflammation, and addiction.

A significant effect of diet was reported in the ANOVA for the expression of tropomyosin receptor kinase B (TrkB), ionized calcium-binding adapter molecule 1 (IBA-1), the phosphorylated and total form of NF-kB, tumoral necrosis factor alpha (TNF-α), and FosB and its truncated variant ΔFosB. The CTRL HFHFD animals showed a significant increase in TrkB, IBA-1, FosB, and ΔFosB and a significant decrease in NF-kB and TNF-α expression when compared to animals on the CTRL STD diet.

A significant effect of treatment was shown in the ANOVA for the expression of brain-derived neurotrophic factor (BDNF), TrkB, glial fibrillar acidic protein (GFAP), IBA-1, the phosphorylated and total form of NF-kB, FosB, and ΔFosB. No significant effect was reported in the groups HFHFD LIG and HFHFD OLHHA when compared to the CTRL HFHFD group. Conversely, the HFHFD OLHHA+LIG group significantly normalized the expression of BDNF, TrkB, GFAP, IBA-1, NF-kB tot, and ΔFosB to control values (CTRL STD). Moreover, combined treatment significantly reduced FosB and phosphorylated NF-kB expression compared to the CTRL HFHFD.

The details of the ANOVA analysis for each protein are shown in Appendix A, along with the Ponceau and staining for each membrane (Appendix A).

#### 2.2.2. Hippocampus

Figure 4A evidences the effect of diet exposure or treatment administration on the expression of the ECS in the hippocampus.

An effect of the diet was evidenced in the ANOVA analysis for the protein expression of DAGLα, DAGLβ, MAGL, FAAH, and the ratio NAPE-PLD/FAAH. A significant decrease was observed in DAGLβ expression, and a significant increase was noticed in DAGLα, MAGL, FAAH, and NAPE/FAAH ratio when comparing CTRL HFHFD animals versus CTRL STD animals.

The ANOVA displayed an effect of the treatment on the protein expression of CB1, DAGLβ, AMGL, NAPE-PLD, FAAH, and the ratio synthesis/degradation of endocannabinoids. The administration of all the treatments significantly reduced the MAGL expression to values comparable to the CTRL STD group. Treatments with OLHHA or a combination of OLHHA+LIG significantly increased NAPE-PLD expression and reduced the NAPE-PLD/FAAH ratio compared to both the CTRL HFHFD and HFHFD LIG groups. On the other hand, OLHHA treatment alone significantly reduced CB1 and increased DAGLβ expression compared to the CTRL HFHFD group. Differently from the other treatments, OLHHA+LIG significantly increased the expression of FAAH and the ratios DAGLα/MAGL and DAGLβ/MAGL in a significant manner when compared to the CTRL HFHFD.

Figure 4B evidences the effect of diet exposure or treatment administration on the expression of proteins regulating food intake.

No effect of diet was shown in ANOVA analyses.

Conversely, treatment administration showed a significant effect on the expression of GLP1-R and MOR. The HFHFD OLHHA+LIG group reported a significant increase in GLP-1R and mu-opioid receptor (MOR) expression when compared to the CTRL HFHFD.

Figure 5A shows the effect of diet exposure and treatment administration on proteins partaking in the insulin pathway and its activation/inactivation by phosphorylation.

The ANOVA reported an effect of diet on the protein expression of IRβ, IRS-1, and its phosphorylated forms in TYR and serine (SER), the phosphorylated form of PI3K. Compared to the CTRL STD, CTRL HFHFD animals reported significant reductions in the expression of IRβ, IRS-1, and the phosphorylated form of PI3K, whereas significant increases were detected in the phosphorylated form of IRS-1, both in SER and TYR.

A significant effect of treatment was found by ANOVA for the expression of total IRS-1 and phosphorylated form in TYR. However, no significant differences were highlighted among the HFHFD LIG and HFHFD OLHHA groups versus the CTRL HFHFD. Conversely, the HFHFD OLHHA+LIG significantly reduced the IRS-1 phosphorylation at TYR compared to the CTRL HFHFD and increased IRS-1 total expression compared to HFHFD LIG and HFHFD OLHHA groups.

Figure 5B displays the effect of diet exposure and treatment administration on metabolic sensors and their activation/inactivation by phosphorylation.

Based on the ANOVA, diet had a significant effect on the phosphorylation of mTOR and the total and phosphorylated form of ERK2. CTRL HFHFD rats showed a significant decrease in mTOR phosphorylation and significant alterations in ERK2 total and phosphorylated forms when compared to CTRL STD rats.

Two-way ANOVA evidenced an effect of the treatment on the expression of total mTOR and phosphorylated ERK2. The administration of all treatments in HFHFD-fed rats resulted in the increased expression of phosphorylated ERK1 versus the CTRL HFHFD. Only treatment with OLHHA or OLHHA+LIG caused a significant increase in ERK2 phosphorylation when compared with the CTRL HFHFD group.

Figure 6A shows variations elicited by diet or treatment in total tau, its phosphorylated forms AT8 and AT100, and the expression of a kinase responsible for phosphorylating tau, CDK5.

In the ANOVA, the diet effect was significant for the expression of total TAU, CDK5, and p35. Compared with CTRL STD rats, CTRL HFHFD animals displayed a significant increase in tau total protein and p35, and a significant decrease in CDK5.

Two-way ANOVA reported an effect of treatment for the expression of total tau, CDK5, p35, and p25. The HFHFD OLHHA group displayed a significant decrease in CDK5 and p35 expression when compared to vehicle administration (CTRL HFHFD). The combined treatment of OLHHA+LIG significantly reduced the expression of AT8 and p25 when compared to the HFHFD OLHHA group. In the OLHHA+LIG group, a significant reduction in total tau and CDK5 was also noticed versus the CTRL HFHFD group. Conversely, a significant increase was shown in the expression of AT100 when compared to the CTRL HFHFD.

Figure 6B shows variations elicited by diet or treatment in the markers of trophism, neuroinflammation, and addiction.

Two-way ANOVA detected a significant effect of diet in the expression of TrkB, GFAP, phosphorylated NF-kB, TNFα, FosB, and ΔFosB. The CTRL HFHFD rats showed an increased expression of GFAP and IBA-1 and a significantly higher phosphorylation of NF-kB when compared to the CTRL STD. Also, they displayed significantly reduced TNF-α and Δ FosB.

Two-way ANOVA detected a significant effect of treatment on GFAP, phosphorylated NF-kB, and TNFα. The HFHFD OLHHA group presented a significantly reduced expression of GFAP and phosphorylated NF-kB when compared to the CTRL HFHFD. No significant effect was reported for OLHHA+LIG administration, whereas LIG successfully reduced GFAP expression when compared to the CTRL HFHFD.

The details of the ANOVA analysis for each protein are shown in Appendix A, along with the Ponceau and staining for each membrane (Appendix A).

#### 2.2.3. Prefrontal Cortex

Figure 7A shows the alterations detected in the expression of the ECS due to exposure to diet or treatment administration.

Two-way ANOVA evidenced an effect of the diet on the expression of CB2, PPARα, DAGLα, DAGLβ, MAGL, NAPE-PLD, FAAH, and the ratios of the synthesis and degradation of endocannabinoids. The CTRL HFHFD rats displayed a significant decrease in the expression of CB2, DAGLα, and NAPE-PLD and significant increases in PPARα, DAGLβ, MAGL, FAAH, and the ratios DAGLα/MAGL, DAGLβ/MAGL, and NAPE/FAAH when compared to the CTRL STD.

Similarly, the effect of treatment was detected by two-way ANOVA for the expression of CB2, PPARα, DAGLα, DAGLβ, MAGL, FAAH, and the ratios of the synthesis and degradation of endocannabinoids. No significant effect was noticed in HFHFD LIG rats. On the other hand, the HFHFD OLHHA group presented a significant increase in the expression of CB2 and DAGLα and a significant decrease in PPARα protein expression, compared to the CTRL HFHFD. Combined treatment with OLHHA+LIG significantly ameliorated PPARα, DAGLα, DAGLβ, and MAGL expression when compared to the CTRL HFHFD group but also to the HFHFD LIG and HFHFD OLHHA groups.

Figure 7B evidences the effect of diet exposure or treatment administration on the expression of proteins regulating food intake.

A significant effect of the diet was noticed in the two-way ANOVA analysis for the expression of MOR: CTRL HFHFD animals showed significantly decreased MOR expression when compared to CTRL STD ones.

Conversely, ANOVA evidenced a significant effect for treatment on the expression of GLP1-R, MOR, and NPY1-R. HFHFD OLHHA+LIG rats presented significantly increased GLP-1R expression when compared to vehicle administration (CTRL HFHFD). A significant reduction in the expression of MOR was also noticed in HFHFD LIG animals compared to CTRL HFHFD animals.

As shown in Figure 8A, significant variations were reported in the expression of proteins partaking in the insulin pathway due to diet or treatment.

A significant effect of diet was shown in the two-way ANOVA for the expression of total IRS-1 and its phosphorylated SER form, the phosphorylated forms of PI3K and AKT, and total GSK3β. CTRL HFHFD rats displayed significant decreases in the expression of total IRS-1 and phosphorylated forms of IRS-1 (SER), PI3K, and AKT compared to the CTRL STD group.

The effect of treatment was reported by two-way ANOVA for the expression of IRβ, IRS-1, and its phosphorylated form in SER, the phosphorylated form of AKT, and total GSK3β. Treatment with OLHHA in HFHFD rats significantly restored the expression of phosphorylated IRS-1 (SER) versus the CTRL HFHFD group. Treatment with OLHHA+LIG significantly re-established the expression of phosphorylated IRS-1 (SER), phosphorylated AKT, and total GSK3β while reducing IRS-1 and AKT total expression. These effects were significant not only when compared to the CTRL HFHFD group but also with LIG HFHFD and OLHHA HFHFD groups. No effect was shown in LIG-administered rats.

Figure 8B shows significant variations occurring in the protein expression of metabolic sensors and their activation by phosphorylation in the PFC due to diet or treatment.

A significant effect of the diet was reported by two-way ANOVA for the expression of mTOR, total ERK1 and its phosphorylated form, and ERK2. CTRL HFHFD rats showed a significantly lower expression of total mTOR and increased total ERK1, phosphorylated ERK1, and total ERK2 when compared with the CTRL STD.

Two-way ANOVA evidenced a significant effect of treatment for the expression of mTOR, total ERK1 and its phosphorylated form, and ERK2. All the treatments (LIG, OLHHA, and OLHHA+LIG) significantly reduced phosphorylated ERK1 expression compared to the CTRL HFHFD. The HFHFD OLHHA+LIG group also presented a significant reduction in total ERK1 and ERK2 compared to the CTRL HFHFD.

Figure 9A shows the effect of diet exposure or treatment administration on the expression of tau, its phosphorylated forms, and its phosphorylating kinase CDK5.

The effect of diet was evidenced by the ANOVA for the expression of AT8, total tau, CDK5, and p25. The CTRL HFHFD group showed a significant increase in AT8 and CDK5, along with a significant decrease in total tau and p25 when compared to the CTRL STD.

The ANOVA revealed an effect of treatment on the protein expression of AT8, CDK5, and p35. Compared to CTRL HFHFD rats, combined treatment with OLHHA+LIG significantly decreased AT8 expression and increased total tau and p35, re-establishing values of expression comparable to the controls.

Figure 9B shows the effect of diet and treatment on the expression of the markers of trophism, neuroinflammation, and addiction.

A significant effect of diet was noticed in the two-way ANOVA for the expression of GFAP, IBA-1, TNFα, and FosB. Compared to the CTRL STD, the CTRL HFHFD group presented significant increases in the levels of GFAP and a reduction in TNF-α. Also, the same animals presented a significant reduction in FosB expression.

The effect of treatment was noticed in the two-way ANOVA for IBA-1 expression: combined treatment with OLHHA+LIG significantly reduced IBA-1 expression when compared to vehicle administration (CTRL HFHFD).

The details of the ANOVA analysis for each protein are shown in Appendix A, along with the Ponceau and staining for each membrane (Appendix A).

### 2.3. Pearson Correlation Analysis

A Pearson Correlation Analysis was performed to determine whether peripheral parameters can be related to protein expression in the brain. Results show that BW, triglycerides, cholesterol, LDL, HDL, ALT, AST, and bilirubin present regio-specific correlations with protein expression in the brain. Table 2, Table 3 and Table 4 show the correlations evidenced and *p* values.

## 3. Discussion

With this study, we demonstrated the efficacy of a combined multitarget treatment approach in ameliorating the peripheral and central metabolic alterations induced by a high-fructose high-fat diet (HFHFD) which leads to complicated obesity. Rats exposed to a STD or HFHFD diet were studied to identify the phenotypic alterations induced by the HFHF diet, analyzing changes in BW, biochemical plasma parameters, and the protein expression of key homeostatic regulatory signaling systems in various brain areas related to food intake regulation and cognition, such as the HYPO, HIPPO, and PFC. In parallel, the effect of the chronic administration of the GLP-1R agonist LIG and the PPARα agonist/CB1 antagonist OLHHA, and their combination was studied. Currently, multiple studies are investigating the combined action of drugs targeting CB1 and PPARα receptors [9], or GLP-1 and CB1 receptors [19], as well as interactions between GLP-1R and PPAR receptors [20,21]. However, no evidence is currently available about treatments combining these three receptors simultaneously.

Several studies have demonstrated the efficacy of LIG for BW reduction, in both clinical and preclinical research [12,13], as well as its ability to improve the peripheral parameters associated with obesity, including blood lipids and hepatic enzymes [22,23]. At the same time, previous research conducted in our laboratory demonstrated that OLHHA administration is effective in reducing BW and metabolic syndrome in obesity models [17].

In this study, a severe obesity model, characterized by the consumption of high-fat foods and exposure to drinking fructose, was used to mimic the conditions of excessive food intake observed in morbid obesity. Our results showed that HFHF consumption increased BW gain, blood lipids, and hepatic enzymes [16]. However, contrary to our expectations, treatment with LIG did not significantly reduce BW, although it improved the plasma lipidic profile, possibly due to the use of a submaximal dose of this GLP-1 agonist. Consistently, previous studies have shown BW reductions with LIG in rats, but at higher doses [24,25,26,27]. On the other hand, in agreement with previous findings, OLHHA reduced BW and plasma bilirubin. Most surprisingly, treatment with the combination of the two drugs was the most effective strategy in this obesity model, significantly reducing BW and improving several plasma biochemical parameters (triglycerides, LDL, ALT, and AST), suggesting that the simultaneous targeting of PPARα, CB1, and GLP-1R receptors may be the key for effective obesity treatment.

One of the first interests of our study was to investigate the expression of the ECS in the various brain regions. In our model, we detected regio-specific alterations to the expression of the ECS in response to diet exposure. In all the analyzed areas, we evidenced significant alterations in the enzymes responsible for endocannabinoid synthesis and degradation, along with disruptions in the ratio synthesis/degradation of both 2-AG and AEA. Endocannabinoid signaling in the HYPO is pivotal for the balance of anorexigenic and orexigenic signals [28], and disruption in the levels of endocannabinoids can be directly related to obesity and metabolic syndrome. Mainly, endocannabinoid action on feeding in the HYPO is mediated by CB1 receptors, which are expressed at both pre-and post-synaptic levels in pro-opiomelanocortin (POMC) neurons (mediating satiety) and NPY/Agouti-related peptide (Agrp) ones (mediating hunger). Disrupted CB1 expression in the HYPO, as observed in animals exposed to a HFHFD, causes the altered modulation of the cross-talk among these neurons, provoking increased hunger signals and altered brain and peripheral metabolism [28]. On the other hand, the ECS in the HIPPO is implied in mediating the hedonic aspects of eating but also in learning and memory: alterations in ECS signaling that we detected in this region are compatible with previous results obtained by Massa et al. [29], showing that a HFHFD is associated with disruptions in endocannabinoids which can alter the important processes of synaptic transmission and plasticity that influence cognition. Similarly, various studies reported ECS alterations in the PFC in response to palatable food consumption and abstinence [30,31,32,33,34]. Disruptions of the ECS in this region can compromise processes of synaptic plasticity and neurotransmitter dynamic balance, resulting in altered cognitive behavior [35,36].

As expected, treatment with LIG in HFHFD-fed rats was not effective in restoring ECS alterations in any brain area. On the other hand, thanks to its activity on CB1 and PPARα receptors, OLHHA was partially effective in recovering alterations noticed in the HYPO, HIPPO, and PFC. However, once more, the combined treatment (OLHHA+LIG) was the most effective in restoring ECS protein expression to levels comparable to standard-fed rats in all the brain areas analyzed, suggesting a synergic action of the two drugs.

The HFHFD induced significant alterations in the expression of cerebral GLP-1R: compatibly with previous studies [37], we detected a significant increase in the GLP-1R expression in the HYPO. Conversely, the same animals reported a significantly lower expression of GLP-1R in the HIPPO and PFC. Moreover, HFHFD-fed animals displayed a significant increase in NPY1-R expression, which may drive overeating [38]. Also, significant alterations were noticed in MOR transmission both in the HYPO and PFC [39], sustaining the fact that a HFHFD induces alterations in reward signaling, which sustains palatable food consumption.

Interestingly, only the combined treatment with OLHHA+LIG significantly increased the expression of GLP-1R in HFHFD-fed animals, enhancing GLP-1 activity in all the analyzed brain areas, suggesting that this receptor activation is fundamental for its pharmacological action. NPY1-R normal expression, which was boosted by the diet, was dramatically reduced only by the combined treatment, thus removing the hyperphagic NPY-dependent signal. This pharmacological action may justify the BW loss noticed in these animals [40]. No similar effect was shown in animals exposed to a HFHFD and treated with LIG or OLHHA alone, evidencing, once more, the positive effects of the combined treatment.

Another goal of our study was to determine whether a HFHFD could result in cerebral insulin resistance. Starting with the HYPO, we noticed signs compatible with insulin resistance in HFHFD-fed animals, which provoked an increased expression of IRβ, chronic activation of PI3K, and decreased activity of GSK3β. Chronic hyperinsulinemia, commonly observed in obesity, can lead to the persistent stimulation of hypothalamic insulin receptors and disrupt normal insulin signaling, causing metabolic dysregulation and appetite imbalances, contributing to overeating and sustaining BW gain [41,42]. Moreover, increased PI3K signaling in the HYPO can derive from leptin resistance, which is common in HFHFD-fed rats [43]. Disrupted GSK3β in HFHFD-fed animals directly depends on aberrant PI3K signaling and compromises whole-body energy balance and glucose metabolism [44]. The HFHFD also provoked changes in the expression of proteins partaking to the insulin pathway in cognitive areas such as the HIPPO and PFC, where we noticed a decreased expression of IRβ and IRS-1 and a decreased activation of PI3K and AKT, leading globally to a decrease in insulin signaling [41], which has been previously associated with a deficit in memory and learning and disrupted food reward signaling [45,46,47,48,49].

Surprisingly, the administration of LIG was not effective in reverting any of the alterations noticed in insulin pathway protein expression; despite LIG being widely used for the therapy of diabetes [14], it had no efficacy in reverting signs of brain insulin resistance in our model. In this case, the use of a submaximal dose should not have an impact because LIG generally has a glucose-regulating action and increases insulin sensitivity at lower doses compared to the ones necessary for BW reduction [15]. Similarly, little to no significant improvements were detected with OLHHA administration. Conversely, the combined administration of the two drugs (OLHHA+LIG) re-established normal IRS-1 signaling in all the analyzed areas, normalized IRβ and PI3K expression in the HYPO, and restored AKT activation and GSK3β expression in the PFC.

Additionally, the downstream proteins involved in insulin pathways, such as mTOR and ERK1/2, were severely disrupted by the HFHFD exposure. In the HYPO, mTOR signaling is fundamental for regulating energy balance, glucose, lipid metabolism, and hepatic function [50,51], and, all over the brain, it has a pivotal role in promoting synaptic function, cell proliferation, and autophagy [52,53,54]. Similarly, ERK1/2 function as metabolic sensors and regulate insulin sensibility and neuronal energy balance [55]. Moreover, ERK is linked to several cognitive functions, including memory, learning, and attention, and dysfunctions in its pathway have been associated with the development of neurodegenerative diseases [56,57]. Disruptions in mTOR signaling provoked by HFHFD exposure were restored only by combined treatment with OLHHA+LIG, whereas ERK1/2 protein expression and activation were normalized by the administration of all treatments (LIG, OLHHA, and OLHHA+LIG).

Knowing the link between obesity and neurodegenerative diseases and the association between brain insulin resistance and tauopathies [58], we analyzed the expression of total tau and its phosphorylated forms, along with the expression of CDK5 and catalytic subunits, p35 and p25, which are involved in the process of tau phosphorylation. The HFHFD changed total tau expression in all the brain areas analyzed and provoked a significant increase in its phosphorylated form, AT8, in both the PFC and HIPPO, suggesting a potential disruption in tau homeostasis and the onset of tau hyperphosphorylation, which could be indicative of early-stage neurodegenerative processes or increased vulnerability to cognitive decline, accordingly to previous evidence [59,60]. Significant alterations in tau observed in the HFHFD were accompanied by a disruption in CDK5 and its catalytic subunits p35 and p25 in all the brain regions analyzed, suggesting the involvement of this pathway in tau phosphorylation [61]. Rising evidence suggests that GLP-1R agonists may act with neuroprotective effects [62,63]; however, in this study, we report no pharmacological activity of LIG in restoring tau or CDK5-associated protein expression. Little to no effect was observed with the administration of OLHHA; however, once more, the combination treatment (OLHHA+LIG) was effective in restoring some of the alterations in the expression of these proteins in all the brain areas analyzed.

As expected, HFHFD exposure resulted in glial and microglial activation [64,65], with increased GFAP in the HIPPO and PFC and increased IBA-1 expression in the HYPO and PFC. As previously published [66], in the hippocampus these alterations were coupled with the activation of NF-kB, suggesting neuroinflammation-based disruptions of the hippocampal function supporting the cognitive alterations associated with obesity. Surprisingly, HFHFD-fed animals display increased BDNF and TrkB expression in the HYPO: knowing that BDNF collaborates in the control of energy and glucose balance, we suppose that its increment is trying to compensate for the aberrant variations induced by the HFHFD, which were discussed previously [67]. Conversely, a reduced expression of TrkB was noticed in the HIPPO, sustaining the hypothesis of impaired neuronal survival in animals fed with a HFHFD [68]. Once more, rats treated with LIG displayed no significant change in protein expression due to pharmacological treatment. Interestingly, changes occurring in the HIPPO were significantly restored only by the OLHHA treatment, whereas alterations due to the HFHFD in the HYPO were successfully restored by the administration of the combination of the two drugs (OLHHA+LIG).

Moreover, we demonstrate that the peripheral parameters present strong correlations with brain protein expression, meaning that peripheral biomarkers could potentially serve as non-invasive indicators of altered neurological processes or brain health status in obese patients, thus favoring special medical interventions to prevent the onset of cognitive impairment.

Finally, with this study, we demonstrated that exposure to a HFHFD for 8 weeks is sufficient to induce BW gain and increase plasma triglycerides, LDL, and hepatic parameters. A HFHFD provoked disruptions in the expression of the proteins regulating food intake, the ECS, insulin pathways, and inflammation and resulted in altered tau expression and phosphorylation in various brain areas, which is indicative of signs of neurodegeneration. Based on our results, the administration of the GLP-1R agonist LIG alone or a double PPARα agonist/CB1 antagonist (OLHHA) with anorexigenic activity is not sufficient to completely revert the alterations noticed, both at the peripheral and central level. On the other hand, the combined treatment with both compounds (OLHHA+LIG) resulted in the most effect in normalizing BW, plasma parameters, and ameliorating aberrant brain protein expression. These findings align with previous research, underlying that CB1 antagonism can potentiate GLP-1R agonist activity [69,70,71,72].

Of course, some limitations must be accounted for in this study: it was developed only in male rats, lacking a genre perspective. Moreover, even though the results obtained in the brain suggest the presence of cognitive impairment in HFHFD animals, no behavioral tests have been conducted to support this hypothesis. Furthermore, more studies may be needed to clarify the molecular mechanism of action justifying the synergy of the OLHHA+LIG administration.

## 4. Materials and Methods

### 4.1. Animal Model

#### 4.1.1. Animal Protocol and Ethics Statement

The study was conducted using 4- to 5-week-old male Wistar Han International Genetic Standard rats (Crl:WI(Han)) supplied by Charles River Laboratories (Barcelona, Spain). The rats were housed and cared for at the Center for Experimentation and Animal Behavior at the University of Malaga, Spain. At the start of the treatment, the animals weighed 296 ± 13 g and were kept individually under controlled conditions, with ambient temperatures ranging from 22 to 25 °C, 75% humidity, and a 12 h light/dark cycle.

All procedures complied with the European Communities Council Directive 2010/63/EU (EC Regulation 86/609/ECC, 24 November 1986) and Spanish national and regional animal experimentation guidelines (Real Decreto 53/2013). The experimental protocols were approved by the University of Malaga’s Ethical Committee for Animals [73]. The research was conducted following the ARRIVE guidelines (Animal Research: Reporting of In Vivo Experiments). Measures were taken to minimize animal suffering and reduce the number of animals used to the necessary minimum. Authorized protocols for animal research were integrated into the approved project (#11/02/2020/012, Consejería de Agricultura, Ganadería, Pesca y Desarrollo Sostenible, Junta de Andalucía, Spain).

#### 4.1.2. Drugs

The compound OLHHA (N-[1-(3,4-dihydroxyphenyl)propan-2-yl]oleamide) was synthesized by the Research and Development Center for Organic Synthesis for the Chemical-Pharmaceutical Industry (Sintefarma, University of Barcelona, Spain) following a previously established method [17]. OLHHA was dissolved in a vehicle solution composed of 5% Tween 80 (Sigma-Aldrich, San Luis, MO, USA, #P4780) in 0.9% NaCl and administered intraperitoneally (ip) at a dose of 3 mg/kg in a volume of 1 mL/kg. A LIG solution (Cayman Chemical, Ann Arbor, MI, USA, #24727) was prepared by dissolving 1 g of the compound in 4 mL of 0.9% NaCl and was administered subcutaneously (sc) at a dose of 25 µg/kg in a volume of 100 µL/kg. Control groups received either the vehicle solution for i.p. administration or 0.9% NaCl for sc administration.

#### 4.1.3. Experimental Groups

To generate the diet-induced obesity in rats, the animals were fed a high-fat diet (HFD) (4.31 kcal/g) (Research Diet, T-58Y1-58126) and provided ad libitum access to a fructose solution (42 g/L in water), following a modified version of the method by [16]. Wistar rats were divided into five groups of eight animals each (n = 8): (1) Negative Control group (CTRL STD), which received a standard rodent diet; (2) Vehicle group (CTRL HFHFD), serving as a negative control for treatments and receiving HFHFD along with the vehicle but no drugs; (3) HFHFD group treated with LIG (HFHFD LIG); (4) HFHFD group treated with OLHHA (HFHFD OLHHA); and (5) HFHFD group treated with a combination of OLHHA and LIG (HFHFD OLHHA+LIG). All groups, except CTRL STD, were provided with fructose-supplemented drinking water for 8 weeks alongside daily treatment administration. The BW was recorded daily.

### 4.2. Plasma Biochemical Analysis

The animals were euthanized by i.p. injection of sodium pentobarbital (50 mg/kg BW) 2 h after receiving their final treatment dose. Blood samples were subsequently collected through cardiac puncture into EDTA-2Na tubes and centrifuged at 2000× *g* for 10 min at 4 °C. Brains were then collected by dissection and stored at −80°C for later analysis.

Plasma levels of triglycerides, total cholesterol, high-density lipoprotein (HDL), low-density lipoprotein (LDL), bilirubin, and hepatic enzymes, including alanine aminotransferase (ALT) and aspartate aminotransferase (AST), were quantified using a Hitachi 737 Automatic Analyzer (Hitachi, Tokyo, Japan).

### 4.3. Western Blot Analysis

Using Paxinos’ brain atlas, brains were dissected into three different regions of interest (HYPO, HIPPO, and PFC), which were weighted to high accuracy. Proteins were extracted from tissue samples on ice using a lysis buffer (50 mM Tris-HCl pH 7.4, 150 mM NaCl, 0.5% NaDOC, 1% Triton ×100, 1 mM EDTA pH 8, and 0.2% SDS) containing a protease and phosphatase inhibitor cocktail. Protein levels were quantified using the Bradford assay, like [74,75]. Equal amounts of protein extract (15 µg) were resolved on 4–12% Bis-Tris Criterion XT Precast Gel (Bio-Rad, Hercules, CA, USA, cat. #3450124) and subsequently transferred to 0.2 µm nitrocellulose membranes (Bio-Rad, Hercules, CA, USA). Membranes were blocked in TBS-T (50 mM Tris-HCl, pH 7.6, 200 mM NaCl, and 0.1% Tween 20) supplemented with 2% bovine serum albumin (BSA, fraction V; Roche, Basel, Switzerland) for 1 h at room temperature. Overnight incubation at 4 °C with specific primary antibodies was conducted to detect target proteins. Appendix A shows antibody details. Following thorough washing in TBS with 1% Tween 20, HRP-conjugated secondary antibodies (anti-rabbit or anti-mouse IgG (H + L); Promega, Madison, WI, USA) diluted at 1:10,000 were applied for 1 h at room temperature. Protein bands were visualized using enhanced chemiluminescence reagents (Western Blotting Luminol Reagent, Bio-Rad), imaged using the Chemi-Doc TM MP System (Bio-Rad), and quantified using ImageJ software version 1.53t (Rasband, W.S., ImageJ, U.S. NIH).

### 4.4. Statistical Analysis

All the data are expressed as MEAN ± SEM. Data were analyzed by two-way ANOVA (factors: diet and treatment) and Tukey’s post hoc. Two-tail Bravais–Pearson correlation tests were performed for each experimental group to correlate different parameters, as in [34]. Statistical significance was set at *p* < 0.05. The software used for graphics and statistics were GraphPad Prism version 9 and SPSS version 22.

## Figures and Tables

**Figure 1 ijms-26-02420-f001:**
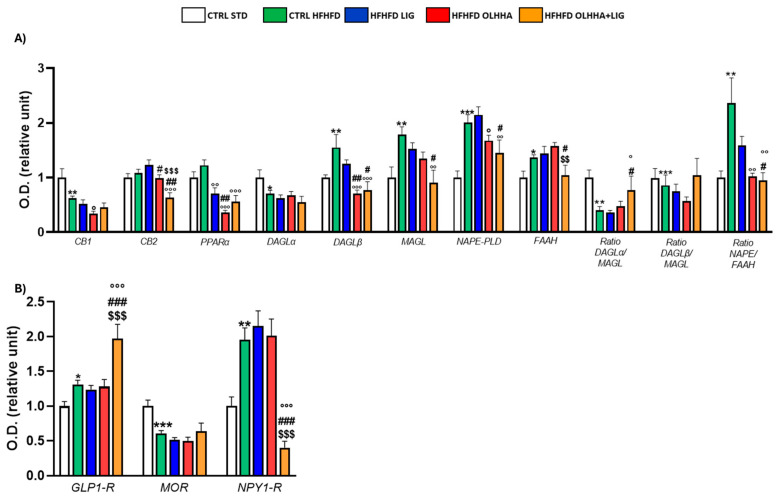
Expression in hypothalamus of proteins partaking to (**A**) endocannabinoid system and (**B**) food intake mediators. Data are expressed as mean ± SEM and analyzed by two-way ANOVA (variables: diet, treatment). * *p* < 0.05, ** *p* < 0.01, *** *p* < 0.001 vs. CTRL STD; ° *p* < 0.05, °° *p* < 0.01, °°° *p* < 0.001 vs. CTRL HFHFD; # *p* < 0.05, ## *p* < 0.01, ### *p* < 0.001 vs. HFHFD LIG; $$ *p* < 0.01, $$$ *p* < 0.001 vs. HFHFD OLHHA.

**Figure 2 ijms-26-02420-f002:**
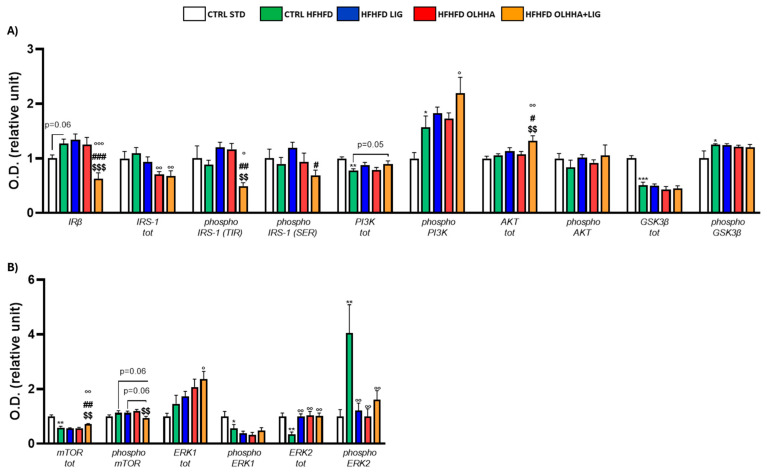
Expression in hypothalamus of proteins partaking to (**A**) insulin pathway and (**B**) metabolic sensors. Data are expressed as mean ± SEM and analyzed by two-way ANOVA (variables: diet, treatment). * *p* < 0.05, ** *p* < 0.01, *** *p* < 0.001 vs. CTRL STD; ° *p* < 0.05, °° *p* < 0.01, °°° *p* < 0.001 vs. CTRL HFHFD; # *p* < 0.05, ## *p* < 0.01, ### *p* < 0.001 vs. HFHFD LIG; $$ *p* < 0.01, $$$ *p* < 0.001 vs. HFHFD OLHHA.

**Figure 3 ijms-26-02420-f003:**
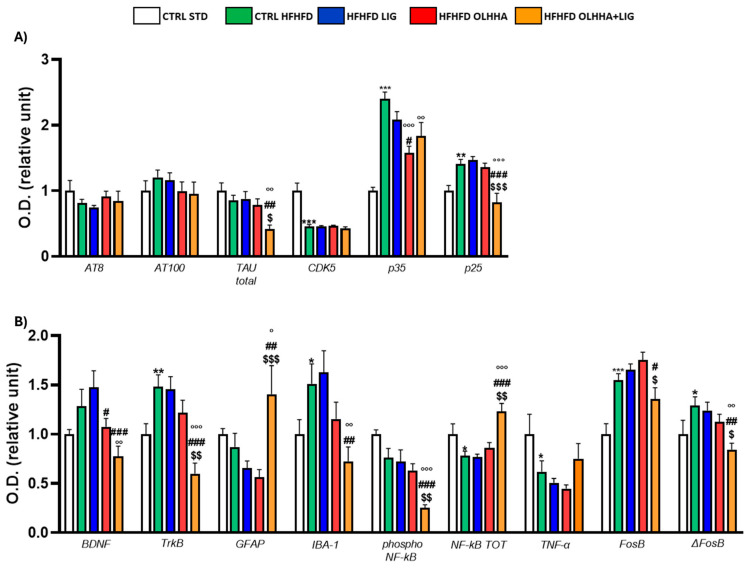
Expression in hypothalamus of proteins partaking to (**A**) tau phosphorylation and (**B**) neurogenesis, neuroinflammation, and addiction. Data are expressed as mean ± SEM and analyzed by two-way ANOVA (variables: diet, treatment). * *p* < 0.05, ** *p* < 0.01, *** *p* < 0.001 vs. CTRL STD; ° *p* < 0.05, °° *p* < 0.01, °°° *p* < 0.001 vs. CTRL HFHFD; # *p* < 0.05, ## *p* < 0.01, ### *p* < 0.001 vs. HFHFD LIG; $ *p* < 0.05, $$ *p* < 0.01, $$$ *p* < 0.001 vs. HFHFD OLHHA.

**Figure 4 ijms-26-02420-f004:**
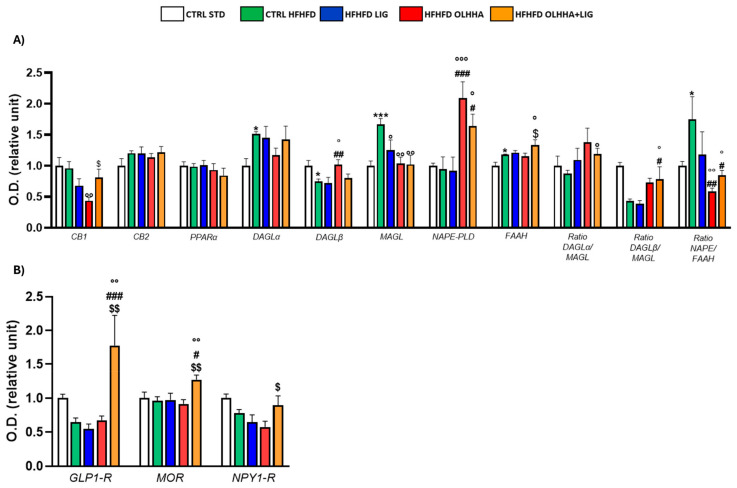
Expression in hippocampus of proteins partaking to (**A**) endocannabinoid system and (**B**) food intake mediators. Data are expressed as mean ± SEM and analyzed by two-way ANOVA (variables: diet, treatment). * *p* < 0.05, *** *p* < 0.001 vs. CTRL STD; ° *p* < 0.05, °° *p* < 0.01, °°° *p* < 0.001 vs. CTRL HFHFD; # *p* < 0.05, ## *p* < 0.01, ### *p* < 0.001 vs. HFHFD LIG; $ *p* < 0.05, $$ *p* < 0.01 vs. HFHFD OLHHA.

**Figure 5 ijms-26-02420-f005:**
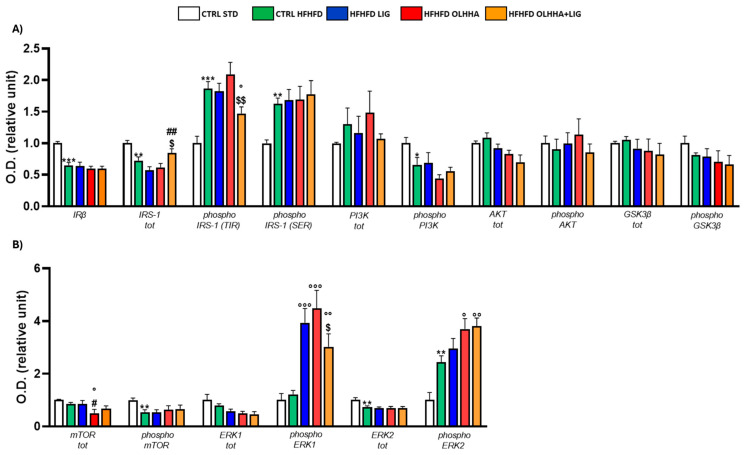
Expression in hippocampus of proteins partaking to (**A**) insulin pathway and (**B**) metabolic sensors. Data are expressed as mean ± SEM and analyzed by two-way ANOVA (variables: diet, treatment). * *p* < 0.05, ** *p* < 0.01, *** *p* < 0.001 vs. CTRL STD; ° *p* < 0.05, °° *p* < 0.01, °°° *p* < 0.001 vs. CTRL HFHFD; # *p* < 0.05, ## *p* < 0.01 vs. HFHFD LIG; $ *p* < 0.05, $$ *p* < 0.01 vs. HFHFD OLHHA.

**Figure 6 ijms-26-02420-f006:**
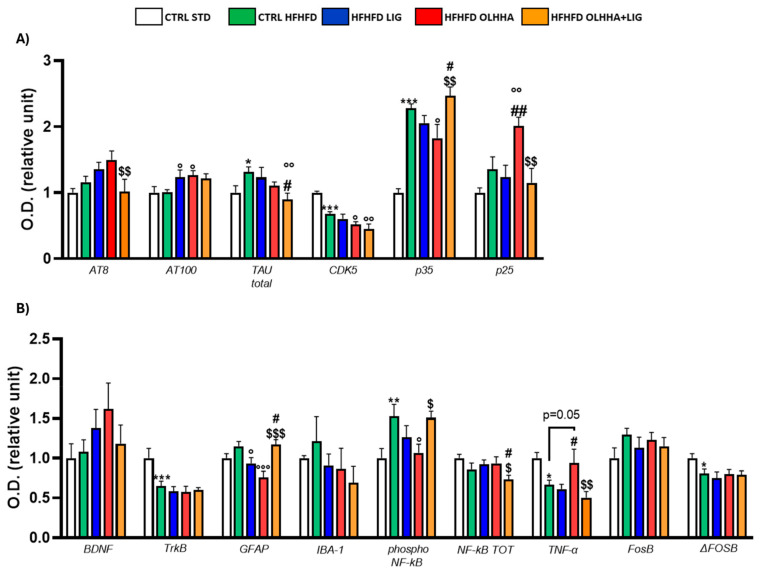
Expression in hippocampus of proteins partaking to (**A**) tau phosphorylation and (**B**) neurogenesis, neuroinflammation, and addiction. Data are expressed as mean ± SEM and analyzed by two-way ANOVA (variables: diet, treatment). * *p* < 0.05, ** *p* < 0.01, *** *p* < 0.001 vs. CTRL STD; ° *p* < 0.05, °° *p* < 0.01, °°° *p* < 0.001 vs. CTRL HFHFD; # *p* < 0.05, ## *p* < 0.01 vs. HFHFD LIG; $ *p* < 0.05, $$ *p* < 0.01, $$$ *p* < 0.001 vs. HFHFD OLHHA.

**Figure 7 ijms-26-02420-f007:**
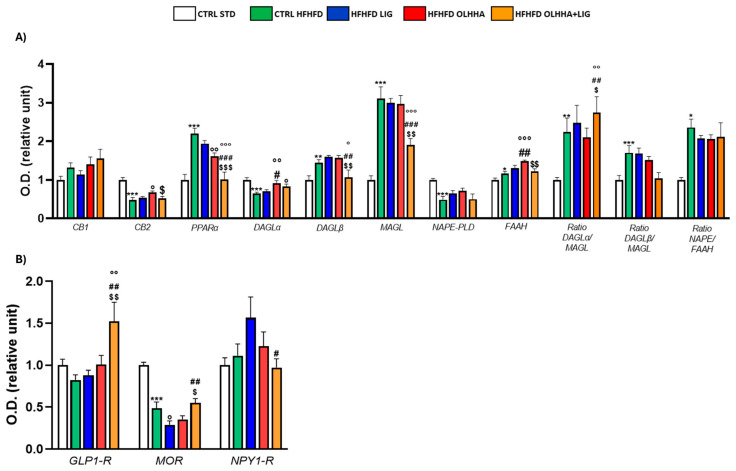
Expression in prefrontal cortex of proteins partaking to (**A**) endocannabinoid system and (**B**) food intake mediators. Data are expressed as mean ± SEM and analyzed by two-way ANOVA (variables: diet, treatment). * *p* < 0.05, ** *p* < 0.01, *** *p* < 0.001 vs. CTRL STD; ° *p* < 0.05, °° *p* < 0.01, °°° *p* < 0.001 vs. CTRL HFHFD; # *p* < 0.05, ## *p* < 0.01, ### *p* < 0.001 vs. HFHFD LIG; $ *p* < 0.05, $$ *p* < 0.01, $$$ *p* < 0.001 vs. HFHFD OLHHA.

**Figure 8 ijms-26-02420-f008:**
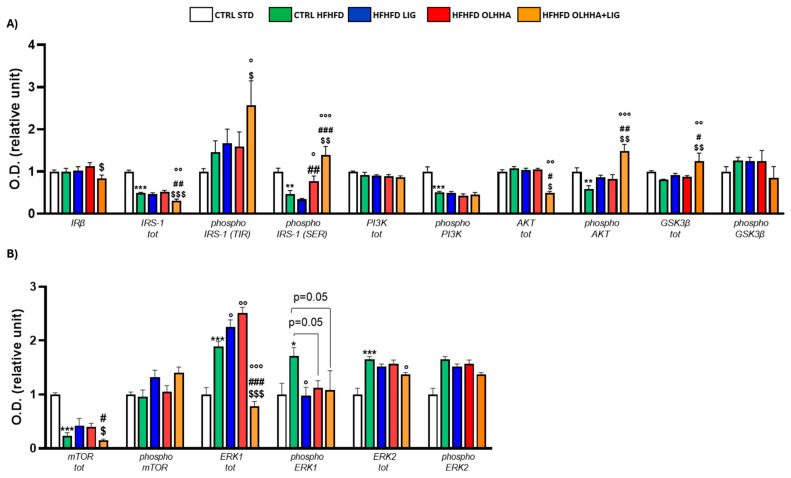
Expression in prefrontal cortex of proteins partaking to (**A**) insulin pathway and (**B**) metabolic sensors. Data are expressed as mean ± SEM and analyzed by two-way ANOVA (variables: diet, treatment). * *p* < 0.05, ** *p* < 0.01, *** *p* < 0.001 vs. CTRL STD; ° *p* < 0.05, °° *p* < 0.01, °°° *p* < 0.001 vs. CTRL HFHFD; # *p* < 0.05, ## *p* < 0.01, ### *p* < 0.001 vs. HFHFD LIG; $ *p* < 0.05, $$ *p* < 0.01, $$$ *p* < 0.001 vs. HFHFD OLHHA.

**Figure 9 ijms-26-02420-f009:**
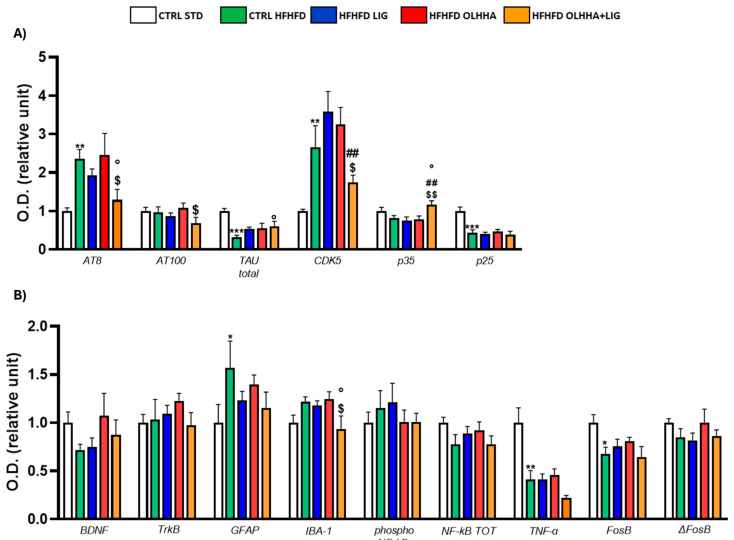
Expression in prefrontal cortex of proteins partaking to (**A**) tau phosphorilation and (**B**) neurogenesis, neuroinflammation, and addiction. Data are expressed as mean ± SEM and analyzed by two-way ANOVA (variables: diet, treatment). * *p* < 0.05, ** *p* < 0.01, *** *p* < 0.001 vs. CTRL STD; ° *p* < 0.05 vs. CTRL HFHFD; ## *p* < 0.01 vs. HFHFD LIG; $ *p* < 0.05, $$ *p* < 0.01 vs. HFHFD OLHHA.

**Table 1 ijms-26-02420-t001:** Body weight gain and plasma biochemical parameters measurements. Data are expressed as mean ± SEM and analyzed by two-way ANOVA (variables: diet, treatment). ** *p* < 0.01, *** *p* < 0.001 vs. CTRL STD; ° *p* < 0.05, °° *p* < 0.01, °°° *p* < 0.001 vs. CTRL HFHFD; # *p* < 0.05, ## *p* < 0.01, ### *p* < 0.001 vs. LIG; $$ *p* < 0.01, $$$ *p* < 0.001 vs. OLHHA.

	CTRL STD	CTRL HFHFD	HFHFD LIG (25 µg/kg)	HFHFD OLHHA (3 mg/kg)	HFHFD OLHHA+LIG (3 mg/kg + 25 µg/kg)
BW gain (mg)	39.130 ± 3.182	165.900 ± 11.160 ***	146.300 ± 8.277	139.400 ± 9.388 °	106.500 ± 4.567 °°°##$$
Triglycerides (mg/dL)	116.9 ± 7.921	152.6 ± 9.795 **	96.80 ± 5.173 °°°	131.3 ± 7.610 #	53.86 ± 6.580 °°°##$$$
Cholesterol (mg/dL)	55.61 ± 2.058	66.88 ± 4.662	72.13 ± 5.034	70.75 ± 4.439	79.75 ± 7.504
HDL (mg/dL)	22.23 ± 2.045	23.60 ± 1.011	31.88 ± 4.741 °	28.00 ± 1.225	23.50 ± 3.246 #
LDL (mg/dL)	44.92 ± 3.102	77.06 ± 8.127 **	44.58 ± 6.084 °°	65.12 ± 5.721 #	36.00 ± 5.672 °°°$$
ALT (IU/L)	40.80 ± 1.800	52.43 ± 1.193	52.71 ± 2.801	50.86 ± 3.038	33.50 ± 6.651 °°##$$
AST (IU/L)	108.8 ± 3.292	123.3 ± 4.448	116.7 ± 6.210	102.9 ± 4.081	73.14 ± 13.27 °°°###$$
Bilirubin (mg/dL)	0.093 ± 0.007	0.188 ± 0.023 **	0.213 ± 0.035	0.126 ± 0.015 ##	0.231 ± 0.027 $$

**Table 2 ijms-26-02420-t002:** Pearson r among peripheral parameters and values of protein expression in hypothalamus. + stands for positive correlation, − for negative one. Only significant values are shown.

Hypothalamus								
Protein	BW	Triglycerides	Cholesterol	LDL	HDL	ALT	AST	Bilirubin
**CB1**		+ (*p* < 0.05)	− (*p* < 0.05)			+ (*p* < 0.0001)	+ (*p* < 0.001)	
**PPARα**							+ (*p* < 0.05)	
**MAGL**	+ (*p* < 0.05)							
**NAPE-PLD**	+ (*p* < 0.05)							
**FAAH**					+ (*p* < 0.05)			
**BDNF**		+ (*p* < 0.05)						
**TrkB**	+ (*p* < 0.01)	+ (*p* < 0.05)				+ (*p* < 0.05)		
**GFAP**		+ (*p* < 0.05)						
**AT8**						+ (*p* < 0.01)		− (*p* < 0.05)
**AT100**						+ (*p* < 0.0001)	+ (*p* < 0.05)	− (*p* < 0.05)
**TAU**								
**CDK5**			− (*p* < 0.05)			+ (*p* < 0.0001)	+ (*p* < 0.01)	− (*p* < 0.01)
**P25**					+ (*p* < 0.05)			
**NF-kB**		− (*p* < 0.01)		− (*p* < 0.05)				
**pNF-kB**		+ (*p* < 0.05)				+ (*p* < 0.01)	+ (*p* < 0.05)	− (*p* < 0.05)
**TNF-α**	− (*p* < 0.05)							
**mTOR**		− (*p* < 0.05)		− (*p* < 0.05)	− (*p* < 0.01)			
**pmTOR**					+ (*p* < 0.05)			
**AKT**			+ (*p* < 0.05)	− (*p* < 0.05)		− (*p* < 0.05)	− (*p* < 0.01)	
**pAKT**			− (*p* < 0.05)					
**pPI3K**		− (*p* < 0.01)				− (*p* < 0.01)	− (*p* < 0.01)	
**GSK3β**	− (*p* < 0.05)		− (*p* < 0.05)			+ (*p* < 0.0001)	+ (*p* < 0.05)	
**pGSK3β**			+ (*p* < 0.05)			− (*p* < 0.0001)	− (*p* < 0.01)	+ (*p* < 0.05)
**pERK1**								+ (*p* < 0.05)
**pERK2**								
**GLP-1R**		− (*p* < 0.01)				− (*p* < 0.01)	− (*p* < 0.01)	
**MOR**	− (*p* < 0.001)							
**NPY1R**	+ (*p* < 0.05)	+ (*p* < 0.05)						
**FOSB**	+ (*p* < 0.01)							
**ΔFOSB**							+ (*p* < 0.05)	

**Table 3 ijms-26-02420-t003:** Pearson r among peripheral parameters and values of protein expression in hippocampus. + stands for positive correlation, − for negative one. Only significant values are shown.

Hippocampus								
Protein	BW	Triglycerides	Cholesterol	LDL	HDL	ALT	AST	Bilirubin
**CB1**			− (*p* < 0.01)			+ (*p* < 0.01)		
**CB2**		− (*p* < 0.05)					− (*p* < 0.05)	+ (*p* < 0.05)
**PPARα**			− (*p* < 0.05)			+ (*p* < 0.05)	+ (*p* < 0.001)	
**MAGL**							+ (*p* < 0.05)	
**BDNF**	+ (*p* < 0.05)							
**TrkB**						+ (*p* < 0.01)		
**GFAP**	− (*p* < 0.05)							
**AT8**					+ (*p* < 0.05)	− (*p* < 0.05)		
**AT100**		− (*p* < 0.05)				− (*p* < 0.05)	− (*p* < 0.05)	+ (*p* < 0.05)
**TAU**		− (*p* < 0.05)					+ (*p* < 0.01)	
**CDK5**						+ (*p* < 0.01)	+ (*p* < 0.001)	
**P35**		− (*p* < 0.05)				+ (*p* < 0.01)	− (*p* < 0.05)	
**P25**								
**NF-kB**						+ (*p* < 0.01)	+ (*p* < 0.05)	
**pNF-kB**						− (*p* < 0.05)		
**IRβ**						+ (*p* < 0.01)	+ (*p* < 0.01)	
**IRS1**					− (*p* < 0.05)			
**AKT**		+ (*p* < 0.05)					+ (*p* < 0.05)	
**pAKT**						+ (*p* < 0.05)		
**PI3K**	− (*p* < 0.05)							
**ERK1**	+ (*p* < 0.05)	+ (*p* < 0.05)				+ (*p* < 0.01)		− (*p* < 0.05)
**pERK1**	− (*p* < 0.0001)							
**ERK2**	+ (*p* < 0.01)					+ (*p* < 0.01)		− (*p* < 0.01)
**pERK2**	− (*p* < 0.0001)							+ (*p* < 0.01)
**GLP-1R**		− (*p* < 0.05)					− (*p* < 0.01)	
**MOR**							− (*p* < 0.05)	
**NPY1R**	− (*p* < 0.05)		− (*p* < 0.05)		− (*p* < 0.01)			
**FOSB**								− (*p* < 0.05)
**ΔFOSB**	− (*p* < 0.01)							

**Table 4 ijms-26-02420-t004:** Pearson r among peripheral parameters and values of protein expression in prefrontal cortex. + stands for positive correlation, − for negative one. Only significant values are shown.

Prefrontal Cortex								
Protein	BW	Triglycerides	Cholesterol	LDL	HDL	ALT	AST	Bilirubin
**CB1**								
**CB2**			− (*p* < 0.05)			+ (*p* < 0.001)	+ (*p* < 0.05)	− (*p* < 0.01)
**PPARα**					+ (*p* < 0.01)			
**MAGL**	+ (*p* < 0.001)							
**NAPE−PLD**						+ (*p* < 0.05)		− (*p* < 0.05)
**FAAH**	+ (*p* < 0.05)					− (*p* < 0.05)		
**GFAP**						− (*p* < 0.05)		+ (*p* < 0.01)
**TAU**						+ (*p* < 0.05)		
**CDK5**	+ (*p* < 0.01)							
**P25**		+ (*p* < 0.05)				+ (*p* < 0.001)	+ (*p* < 0.01)	− (*p* < 0.05)
**TNF-α**	− (*p* < 0.001)							
**IRS1**	− (*p* < 0.05)	+ (*p* < 0.05)				+ (*p* < 0.001)	+ (*p* < 0.01)	
**P** **(tyr)IRS**		− (*p* < 0.05)						
**p(** **ser)IRS1**		− (*p* < 0.05)	− (*p* < 0.01)				− (*p* < 0.05)	
**MTOR**	+ (*p* < 0.01)					+ (*p* < 0.01)	+ (*p* < 0.05)	
**pMTOR**		− (*p* < 0.05)						
**AKT**		+ (*p* < 0.01)		+(*p* < 0.05)			+ (*p* < 0.01)	
**pAKT**	− (*p* < 0.05)	− (*p* < 0.01)	+ (*p* < 0.05)					+ (*p* < 0.01)
**pPI3K**	− (*p* < 0.05)							
**GSK3β**		− (*p* < 0.05)	+ (*p* < 0.01)				+ (*p* < 0.001)	
**ERK1**	+ (*p* < 0.01)							− (*p* < 0.05)
**pERK1**	− (*p* < 0.05)							+ (*p* < 0.05)
**ERK2**	+ (*p* < 0.001)							
**pERK2**	− (*p* < 0.01)							+ (*p* < 0.05)
**GLP−1R**	− (*p* < 0.05)	− (*p* < 0.01)	+ (*p* < 0.05)			− (*p* < 0.01)	− (*p* < 0.01)	
**MOR**						+ (*p* < 0.01)		
**NPY1R**	+ (*p* < 0.05)							

## Data Availability

The data that supports the findings of this study are available from the corresponding authors upon reasonable request.

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
