# Peer review of "A Combined GLP-1/PPARa/CB1-Based Therapy to Restore the Central and Peripheral Metabolic Dysregulation Induced by a High-Fructose High-Fat Diet"

_ijms, 2025, doi:10.3390/ijms26062420_

Round 1
Reviewer 1 Report
Comments and Suggestions for Authors
Ceglia et al have put together a good comprehensive preclinical study on the effects of various receptor agonist drugs to improve diet induced obesity in rats. This is very interesting research and the data is very informative however the manuscript writing can be improved as it is written largely as a narrative and less like a scientific publication.
The abstract list HFD to mean high fat diet but throughout the manuscript, HFD is noted to mean High fructose, high fat diet. Please clarify. Also, it would be good to add actual number and real data with statistics into the abstract.
Very first sentence is vague and awkward. Please rephrase.
Line 73- What is meant by this thigh relationship?
Line 86, Seen these premises should be seeing these premises
Line 95-consider HFHFD instead of HFD
line 112-121- Many of the acronyms are not yet introduced here. Please introduce them first so that the reader knows what then mean.
Line 123-remove first of all
The entire results section is written largely like a narrative and discussion rather than a presentation of the results. Please change to have this section to be read more as a scientific discussion of the results.
Line 166- When coming and we noticed are inappropriate terms to use.
Line 531-Remove to sum up and choose another phrase
Table 1-Remove p=0.06 from ALT and p=0.05 vs. CTRL HFD
Comments on the Quality of English LanguageThe language itself is fine, but there are several phrases that are not appropriate for scientific writing such as "first of all"
Author Response
- Ceglia et al have put together a good comprehensive preclinical study on the effects of various receptor agonist drugs to improve diet induced obesity in rats. This is very interesting research and the data is very informative however the manuscript writing can be improved as it is written largely as a narrative and less like a scientific publication.
We thank the reviewer for the time spent evaluating our manuscript and his precious suggestions. We improved the style of the manuscript and submitted it for English editing.
- The abstract list HFD to mean high fat diet but throughout the manuscript, HFD is noted to mean High fructose, high fat diet. Please clarify. Also, it would be good to add actual number and real data with statistics into the abstract.
We accepted the reviewer's suggestion and used HFHFD as the acronym for high-fat high-fructose diet. The acronym was changed all over the manuscript and in the figures. Due to the limited number of words in the abstract (200) it is not possible to add statistics and/or numbers to it. In the template of the journal, whose structure was used to write the whole paper, there is no reference to the inclusion of statistics in the abstract. Examples of other papers published in the same journal: 10.3390/ijms242317009; 10.3390/ijms25041964
- Very first sentence is vague and awkward. Please rephrase.
Following the reviewer’s suggestion, we changed the first phrase of the article.
- Line 73- What is meant by this thigh relationship?
The phrase was reworded.
- Line 86, Seen these premises should be seeing these premises
Line 86 was changed according to the suggestion of the reviewer.
- Line 95-consider HFHFD instead of HFD
As explicated in point 2, HFD was changed in HFHFD.
- line 112-121- Many of the acronyms are not yet introduced here. Please introduce them first so that the reader knows what then mean.
Acronyms have been introduced in the introduction.
- Line 123-remove first of all
“First of all” was removed.
- The entire results section is written largely like a narrative and discussion rather than a presentation of the results. Please change to have this section to be read more as a scientific discussion of the results.
Following the suggestion of the reviewer, the results section was improved.
- Line 166- When coming and we noticed are inappropriate terms to use.
As suggested, Line 166 was rephrased.
- Line 531-Remove to sum up and choose another phrase
“To sum up” was changed to “finally”
- Table 1-Remove p=0.06 from ALT and p=0.05 vs. CTRL HFD
Following the suggestion of the reviewer, p=0.06 from ALT and p=0.05 vs. CTRL HFD were removed.
Comments on the Quality of English Language
- The language itself is fine, but there are several phrases that are not appropriate for scientific writing such as "first of all"
As expressed in point 1, we improved the style of the manuscript and submitted it for English editing.
Reviewer 2 Report
Comments and Suggestions for Authors
In this manuscript, the authors proposed a multitarget therapy that the combined administration of Oleyl hydroxytyrosol ether (OLHHA) and liraglutide to improve both peripheral and central symptoms in a diet-induced obesity animal model. The combined treatment was more effective in promoting weight loss and ameliorating both central and peripheral alterations induced by a high-fat high-fructose diet in rats than was the administration of liraglutide or OLHHA alone. This manuscript reported an effective strategy for treating obesity and its associated comorbidities.
- OLHHA and liraglutide should be introduced in the section of the introduction.
- There are problems with improper or inconsistent punctuation. “Table 3:” should be “Table 3.”. Incorrect use of punctuation in lines 373 and 377.
- Correct the reference format according to the format required by the journal.
- The author's description of the statistical model differs from the representations in the graphs and tables. Two-way ANOVA analysis, it is suggested to give the P-value of each factor and give whether there is interaction.
- I do not understand why the authors want to directly correlate protein expression with phenotypic indicators.
- Line 143, 172, 195, 223, 249, 270, 318: These sentences are superfluous; the author needs to refer to a graph in the description of the corresponding result.
Author Response
In this manuscript, the authors proposed a multitarget therapy that the combined administration of Oleyl hydroxytyrosol ether (OLHHA) and liraglutide to improve both peripheral and central symptoms in a diet-induced obesity animal model. The combined treatment was more effective in promoting weight loss and ameliorating both central and peripheral alterations induced by a high-fat high-fructose diet in rats than was the administration of liraglutide or OLHHA alone. This manuscript reported an effective strategy for treating obesity and its associated comorbidities.
First, we want to thank the reviewer for the precious time spent evaluating our manuscript and for the helpful suggestions made to improve its quality.
- OLHHA and liraglutide should be introduced in the section of the introduction.
Following the reviewer’s suggestion, small paragraphs about liraglutide and OLHHA were added to the introduction.
- There are problems with improper or inconsistent punctuation. “Table 3:” should be “Table 3.”. Incorrect use of punctuation in lines 373 and 377.
Punctuation was revised throughout the manuscript. “Table 3:” was corrected in “Table 3.”
- Correct the reference format according to the format required by the journal.
References were changed according to the format required by the journal.
- The author's description of the statistical model differs from the representations in the graphs and tables. Two-way ANOVA analysis, it is suggested to give the P-value of each factor and give whether there is interaction.
For data analysis, we used a two-way ANOVA to account for two independent variables: diet (STD/HFHFD) and treatment (CTRL, LIG, OLHHA, LIG+OLHHA). The F and P values for diet and treatment effects are provided in the supplementary material to avoid an extensive description in the results section. Due to the absence of treated standard-fed groups (STD LIG, STD OLHHA, STD LIG+OLHHA) in the experimental design, ANOVA could not compute F or P values for the interaction between the two factors. As a result, no interaction statistics are reported in the results or supplementary material, and post-hoc comparisons between the CTRL STD group and the treated HFHFD groups (HFHFD LIG, HFHFD OLHHA, HFHFD LIG+OLHHA) were not feasible.
- I do not understand why the authors want to directly correlate protein expression with phenotypic indicators.
As explained in lines 110-113, 382–385, and 543–548 of the article, we analyzed the correlation between brain protein expression and phenotypic indicators to explore whether peripheral markers could serve as non-invasive indicators of brain signaling alterations. This approach may provide insights into brain health in obese patients and could potentially aid in preventing cognitive impairment in this population.
- Line 143, 172, 195, 223, 249, 270, 318: These sentences are superfluous; the author needs to refer to a graph in the description of the corresponding result.
As suggested by the reviewer, the sentences were canceled, and the graph was mentioned in the description of the corresponding result.
Round 2
Reviewer 2 Report
Comments and Suggestions for Authors
I have no more questions.